# Cross-Interference of VOCs in SnO_2_-Based NO Sensors

**DOI:** 10.3390/nano13050908

**Published:** 2023-02-28

**Authors:** Renjun Si, Yan Li, Jie Tian, Changshu Tan, Shaofeng Chen, Ming Lei, Feng Xie, Xin Guo, Shunping Zhang

**Affiliations:** 1State Key Laboratory of Material Processing and Die & Mould Technology, Department of Materials Science and Engineering, Huazhong University of Science and Technology, Wuhan 430074, China; 2Shenzhen Power Supply Co., Ltd., Shenzhen 518002, China; 3Nanomaterial and Smart Sensor Research Laboratory, Department of Materials Science and Engineering, Huazhong University of Science and Technology, Luo-Yu Road 1037, Wuhan 430074, China; 4Laboratory of Solid State Ionics, School of Materials Science and Engineering, Huazhong University of Science and Technology, Wuhan 430074, China

**Keywords:** Pt doping, VOCs, NO, cross-interference, selectivity

## Abstract

In this work, we studied the influence of cross-interference effects between VOCs and NO on the performance of SnO_2_ and Pt-SnO_2_-based gas sensors. Sensing films were fabricated by screen printing. The results show that the response of the SnO_2_ sensors to NO under air is higher than that of Pt-SnO_2_, but the response to VOCs is lower than that of Pt-SnO_2_. The Pt-SnO_2_ sensor was significantly more responsive to VOCs in the NO background than in air. In the traditional single-component gas test, the pure SnO_2_ sensor showed good selectivity to VOCs and NO at 300 °C and 150 °C, respectively. Loading noble metal Pt improved the sensitivity to VOCs at high temperature, but also significantly increased the interference to NO sensitivity at low temperature. The explanation for this phenomenon is that the noble metal Pt can catalyze the reaction between NO and VOCs to generate more O^−^, which further promotes the adsorption of VOCs. Therefore, selectivity cannot be simply determined by single-component gas testing alone. Mutual interference between mixed gases needs to be taken into account.

## 1. Introduction

Volatile organic compounds (VOCs) and nitrogen oxides (NO_x_), are common toxic and harmful gases. They have short-term and long-term adverse effects on health, including on respiration, the nervous system and the endocrine system [1,2]. In addition, by detecting the concentration of VOCs and NO, a variety of diseases can be discriminated, such as type 1 diabetes mellitus (T1DM) and asthma [3,4]. Therefore, the demand for gas sensors with high accuracy and selectivity is increasing.

There have been a variety of sensors designed and manufactured to detect the concentration of NO_x_ and VOCs. As one of the widely researched chemiresistive sensors, the one adopting a metal oxide semiconductor (MOS) as the sensing material, has received increased attention due to its low cost, high stability, simple operation principles and easy fabrication [5,6]. The most widely used MOS sensor in China is the SnO_2_-based sensor [7,8,9]. Many other promising MOS sensing materials, such as In_2_O_3_ [10,11], ZnO [12,13], a-Fe_2_O_3_ [14,15], WO_3_ [16,17], NiO [18,19] and so on, have been continuously reported for the use of detecting VOCs and NO.

However, the operation of most MOS sensors shows a major disadvantage, poor selectivity [20], which hinders the practical application. This means that any active gas may cause the appearance of sensor response, that is, change of resistance/conductivity of the gas sensitive layer. There have been many reports on improving MOS selectivity, including using aliovalent doping [21,22], a two nanomaterial composite [23,24], noble metal loading [25,26] and so on. By adjusting the electron depletion layer on the surface of materials, constructing potential barriers, adjusting the carrier concentration of materials, changing the distribution of oxygen components on the surface and introducing catalysts, the sensitivity and selectivity of the gas-sensitive material can be improved [27,28]. For example, Wenwen Zeng et al. synthesized hierarchical SnO_2_-Sn_3_O_4_ heterostructural gas sensors; at an optimal operating temperature 150 °C, all the sensors had better responses to NO_2_ than to the other contrast gases (acetone, toluene, xylene, etc.). Its response to 5 ppm NO_2_ was 240, but the response to other contrast VOC gases was less than 6 [29]. Aran Koo et al. prepared Pt-decorated Al-doped ZnO (Pt-AZO) nanoparticles. At an optimal operating temperature of 450 °C, the Pt-AZO showed remarkably enhanced sensing properties, with a sensing response of 421 under exposure to 10 ppm acetone, but almost no response to 10 ppm NO_x_ under the same conditions [30].

In addition, the description and evaluation of sensor selectivity is also critical. At a certain working temperature, the use of one semiconducting gas sensor insufficient to detect contaminated air quality. It can be seen from previous research that, the optimal operating temperature of NO_x_ is mainly at low temperatures, below 250 °C, and the optimal working temperature of VOCs is mainly at high temperatures above 300 °C [31,32,33]. Though it seems simple to create a gas sensor that is highly selective for NO_x_ and VOCs, it is not. Tomoda Munenobu et al. [34] discovered that when NO_2_ exists in the air, the sensitivity to VOCs of TGS2600 (SnO_2_-based) and SmFeO_3_-based gas sensors was enhanced. VOCs may have a much greater impact on the response of NO_x_ at low temperatures. Therefore, the gas interference between VOC and NO_x_ is inevitable. It is necessary to use at least two gas sensors working at different temperatures for the prediction of each of the contaminated levels of VOCs and NO_2_. In practice, in the metal oxide sensor studies, the problem of interference of various gases is not well considered. Only a limited number of articles are devoted to this topic [35,36,37,38,39].

Taking into account the above discussion, it was decided to carry out a project to study the response of SnO_2_-based sensors to VOCs and NO pairs under different conditions and the cross interference between them. Firstly, the gas sensitivity of SnO_2_ and Pt-SnO_2_ to different VOCs and 1ppm NO in the air is characterized. Then, the response curves of VOCs at low temperature with and without Pt in the NO background are further compared. The selectivity in complex environments is redefined by analyzing the difference of sensor response to NO and VOCs under different situations. In addition, the mechanism behind this phenomenon is explained in detail. Finally, it is pointed out that the selectivity cannot be determined simply through the single component gas test, and the interference between the gas mixtures should be considered.

## 2. Materials and Methods

### 2.1. Materials Synthesis

SnO_2_ nanoparticle nanomaterials were firstly synthesized by hydrothermal method. SnCl_4_·5H_2_O and H₂PtCl₆ were purchased from Aladdin Co., Ltd. (Shanghai, China). They were analytical grade and could be used without further purification.

SnCl_4_·5H_2_O (6 mmol) was added into a beaker containing 50 mL benzyl alcohol and stirred at room temperature to form a pale white solution. When the solution was clear, the mixture was transferred to a 100 mL Teflon-lined stainless steel autoclave and reacted at 200 °C for 48 h. After naturally cooling down to room temperature, the obtained precipitate was washed several times with chloroform and anhydrous ethanol. Eventually, the precipitate was dried at 60 °C for 8 h to obtain SnO_2_ nanoparticles.

A certain amount of noble metal ion solution was added into 18 mL of oxide suspension (containing 0.2 g of SnO_2_ powder), stirred vigorously and 2 mL of sodium borohydride (0.5 wt% concentration, in which 5 wt% NaOH was added to create an alkaline environment to slow down the decomposition of sodium borohydride) was dropped in. The suspension slowly changes changed color. It was stirred for 4 h, then transferred to a centrifuge tube, centrifuged at 5000 r/min for 5 min, repeated for three times, and dried at 60 °C to obtain Pt-SnO_2_ nanoparticles.

### 2.2. Characterization

SnO_2_ nanoparticles were analyzed by X-ray diffraction (XRD) performed on a Philips X’Pert diffractometer (2θ from 20° to 80°, λ = 1.5406 Å) with Cu Kα radiation. Micromorphology and element distribution of pure SnO_2_ and Pt-decorated SnO_2_ were characterized by field emission scanning electron microscopy (FE-SEM, Hitachi S-4800, Tokyo, Japan) and an energy dispersive spectrometer (EDS, Hitachi S-4800, Tokyo, Japan). TEM images were obtained by a JEOL 2100F microscope (Tokyo, Japan). The X-ray photoelectron spectroscopy (XPS) test was conducted by a Kratos XSAM800 spectrometer (Manchester, England).

### 2.3. Gas Sensor Fabrication and Measurement

The manufacturing method of the SnO_2_ and Pt-decorated SnO_2_ gas sensors were as follows: 1 mol% H_2_PtCl_6_ and 10 mmol SnO_2_ nanoparticles were dispersed in 40 mL ethanol and magnetically stirred for one hour at room temperature. Then the mixed solution was dried in an oven at 70 °C (pure SnO_2_ gas sensor adds SnO_2_ nanoparticles only). After that, the pre-dried nanoparticles were ground with homemade organic solvent (terpinol 50%, butylcarbitol acetate 30%, dibutyl phthalate 10%, ethyl cellulose 5%, Sorbitan Trioleate 3.5%, 1,4-butyllactone 1% and hydrogenated castor oil 0.5%) and made into pastes by the ball-milling technique. The pastes were printed onto the substrate with pre-fabricated Pt interdigital electrodes by screen printing. The sample was obtained by annealing the synthesized pastes in this experiment at 350 °C for 2 h and then 550 °C for 2 h to evaporate the organic solvent. The mechanical bond between the film particles was improved at the same time by the heat treatment process [30]. Finally, substrates with sensing film were made into sensors.

The gas sensing properties test was implemented using a self-made high-throughput testing platform. The error rate was less than 5% between 100 Ω and 1 GΩ and less than 15% between 1 GΩ and 10 GΩ [32]. In this work, the gas sensing test proceeding in air and 1ppm NO (air balance) was termed the single-component gas test and mutil-component gas test, respectively. The definition of the response was different in reducing and oxidizing gas: for oxidizing gases, the response of the gas sensor is indicated as the quotient of the resistance in target gas divided by the resistance in dry air (R_gas_/R_air_). For the reducing gas, the response is defined as the reciprocal of the response of the oxidizing gas (R_air_/R_gas_).

All sensors were aged at 350 °C for two days to improve the signal stability before measurement of sensing performance. In the temperature range from 100 °C to 350 °C, the response state of the sensor to 10 ppm of different volatile organic compounds, 1 ppm of nitric oxide and 1 ppm of nitrogen dioxide was continuously tested (the temperature interval was 50 °C), in order to obtain the optimum response temperature. Subsequently, at the optimum operating temperature of NO, the responses of VOCs with different concentrations in air and 1 ppm NO (air balance) atmosphere were tested. The interference of volatile organic compounds on the sensitivity of nitrogen oxides at low temperature can be studied according to the results.

## 3. Results and Discussion

### 3.1. Structure Characterization

The XRD patterns of synthesized SnO_2_ are exhibited in Figure 1; all peaks matched perfectly with the cassiterite SnO_2_ phase (JCPDS No. 99-0024) and the SnO_2_ made by the non-aqueous sol-gel method was highly crystalline.

Figure 2a shows the SEM and TEM images of SnO_2_ powder. As can be seen, the structure of SnO_2_ appears to be the accumulation of numerous uniform nanoparticles. To further confirm this feature, high-resolution TEM images of the SnO_2_ powder were taken and the result is displayed in Figure 2b. It can be observed in the TEM images that some areas are clearly distinguished between light and dark, which proves the stacked structure of nanoparticles; the diameter of synthesized SnO_2_ nanoparticles was approximately 20 nm. EDS mapping of Pt-decorated SnO_2_ sensing film is shown in Figure 2c–f. The results evidently show that corresponding signals of O, Sn and Pt were evenly distributed.

In order to explore the mixed valence in Pt-SnO_2_, XPS measurement was done. It was processed by XPSPEAK41 software (Version No.4.1, V G Scientific Ltd., St Leonards, England). The fitting process included establishing background baseline, adding peaks and fitting. From Figure 3, one can clearly see that besides added noble metal and metal oxides, the sample also contains a small amount of carbon. This is because a small amount of carbon may diffuse into the metal oxide at high temperature. The XPS spectrum of Pt was placed at the upper right corner of the survey spectrum; the Pt 4f spectrum can be fitted by two Gaussian peaks, indicating that 4f_5/2_ and 4f_7/2_ exist in Pt-SnO_2_, and the peaks are located at about 74.5 eV and 71.1 eV.

### 3.2. Selective Redefinition

The responses of SnO_2_ and Pt-SnO_2_ to VOCs and NO_x_ at various working temperatures are shown in Figure 4a,b, respectively. It can be seen that the optimal operating temperature of NO and NO_2_ was at 150 °C and 100 °C, and the optimal working temperature of VOCs was mainly at 300 °C. Loading of noble metal Pt improved the sensitivity to VOCs at high temperature. In order to reveal the selectivity of NO_x_ and VOCs, Figure 4c,d displayed the response of pure SnO_2_ and Pt-SnO_2_ to VOCs and NO_x_; the temperatures are 150 °C and 300 °C, respectively. It can be clearly seen that both sensors have good selectivity for VOCs at 300 °C and good selectivity for NO_x_ at 150 °C in the traditional single-component gas test. The selectivity in the multi-component gas environment was next analyzed.

Dynamic resistance curves of SnO_2_ to 1 ppm NO and 10 ppm C_2_H_5_OH at 150 °C are shown in Figure 5a. Figure 5b shows the response of SnO_2_ to 10 ppm ethanol in air and 1 ppm NO (air balance). As one can see, the response of SnO_2_ to 10 ppm ethanol in air was low at 150 °C and in the presence of 1 ppm NO (air balance), the response of 10 ppm ethanol was barely improved. This means that ethanol does not interfere with the response of SnO_2_ to NO at low temperature. Figure 5b shows the response of Pt-SnO_2_ to 1 ppm NO (air balance) was 218 and the response to 10 ppm ethanol was only 3.2, but in the presence of 1ppm NO (air balance), the response of 10 ppm ethanol was significantly improved. The results show that, although Pt-SnO_2_ showed high selectivity to NO at low temperature in a traditional single-component gas test, ethanol does in fact interfere with the response of NO significantly. These results were assigned to the loaded Pt, which catalyzed the product of reaction between NO and ethanol to further react with ethanol.

In order to show the response of ethanol under the multi-component gas test more clearly, the response was defined, which is shown in Figure 6. In the traditional single-component gas test, the response of ethanol was defined as R_air_/R_ethanol_, where R_air_ was the resistance when sensors were in air, and R_ethanol_ was the resistance when sensors were exposed in the ethanol gases, as shown in Figure 6a. The response of ethanol in 1 ppm NO (air balance) was defined as R_NO_/R_NO+ethanol_, which is shown in Figure 6b, where R_NO_ was the resistance of the sensors in 1 ppm NO (air balance) and R_NO+ethanol_ was the resistance of the sensors exposed in 1 ppm NO (air balance) and in 10 ppm ethanol mixed gases.

According to the definition above, the response of SnO_2_ and Pt-SnO_2_ to 10 ppm different VOCs at 150 °C in air and in 1 ppm NO (air balance) was tested, and the results were are shown in Figure 7a,b. The comparison showed that at 150 °C, in the traditional single-component gas test, SnO_2_ and Pt-SnO_2_ had almost no response to different VOCs under the air background. Under the multi-component gas test, the responses of SnO_2_ to 10 ppm VOCs were barely improved, but the responses of Pt-SnO_2_ to different VOCs were greatly improved. This indicated that not only ethanol but also other VOCs interfere with the NO response of Pt-SnO_2_ at low temperature.

In addition, we considered the existence of humidity in practical application scenarios. At 50% RH, the response of SnO_2_ and Pt-SnO_2_ to 10 ppm different VOCs at 150 °C in air and in 1 ppm NO (air balance) was tested, and the results are shown in Figure 7c,d. The results showed that when humidity was present, the conclusion was the same as above, but the sensitivity of SnO_2_ and Pt-SnO_2_ sensors to all gases showed a slightly decreased trend.

Based on the above results, it can be summarized that the responses of VOCs will rise in NO, which in turn decreases the selectivity to NO. Therefore, we cannot simply define the selectivity to NO as a sensitivity difference between NO and VOCs at a certain concentrations in the air background; the interference between gases must be taken into account. In addition, this also implies that gas response-gas concentration curves in the air background, as a method to display sensitivity and selectivity to target gas, are insufficient to demonstrate the selectivity to NO with VOCs present.

It can be summarized that in the single-component gas test, SnO_2_ and Pt-SnO_2_ had good selectivity to NO at 150 °C, the response was proportional to the concentration. Under the background of 1ppm NO (air balance), the response of SnO_2_ to VOCs was not greatly improved, though and there was still a good selectivity for NO; but the response of Pt-SnO_2_ to VOCs was greatly improved, which presented great interference on the selectivity of NO.

### 3.3. Gas-Sensing Mechanism

First, according to the ionosorption model [33], oxygen in the air will diffuse to the surface of the sensing membrane when heated. The adsorbed oxygen molecules in this process will extract electrons from the conduction band and capture the electrons on the surface as ions to form various oxygen ions (O^X−^), including O_2_^−^, O^−^ and O^2−^ which cause the transition of an electron from the top of it.

As indicated in literature [31], O_2_^−^, O^−^ and O^2−^ are stable at 50–200 °C, 200–400 °C and 400–600 °C, respectively. During the reaction, the electron concentration of the sensing film will decrease because of the electronic transition between the top of the conduction band and the surface chemisorption oxygen. This process results in the generation of an electron depletion layer (EDL), which in turn increases the resistance and reduces the conductivity of SnO_2_.

When the metal oxide sensing film is exposed to VOC gases, these gases react with the surface chemisorption oxygen. Without exception, these chemical processes include electrons releasing back to the top of the conduction band. Reactions between surface chemisorption oxygen and VOC gases follow the reaction path (taking ethanol as an example, Equations (1)–(3)) [35,40,41,42,43]:(1)$$C_{2}H_{6}O_{\left(g\right)}+{3O}_{2(\mathrm{ads})}^{-}\to 2{\mathrm{CO}}_{2(g)}+3H_{2}O+3e^{-}$$
(2)$$C_{2}H_{6}O_{\left(g\right)}+{6O}_{\left(\mathrm{ads}\right)}^{-}\to 2{\mathrm{CO}}_{2(g)}+3H_{2}O+{6e}^{-}$$
(3)$$C_{2}H_{6}O_{\left(g\right)}+{6O}_{\left(\mathrm{ads}\right)}^{2-}\to 2{\mathrm{CO}}_{2(g)}+3H_{2}O+12e^{-}$$

According to the above analysis, O^−^ ions play a key role in the adsorption of VOCs; O^−^ ions exist between 200–400 °C, so the response of SnO_2_ to VOCs is relatively high at around 300 °C. When the temperature is higher than 400 °C or lower than 200 °C, O^X−^ mainly exists in the form of O^2−^ and O_2_^−^, so the response of the sensor to VOCs decreases rapidly. When the temperature is lower than 200 °C, O_2_^−^ plays a major role in NO adsorption [44]. SnO_2_ shows good sensitivity to NO at lower temperatures. Therefore, in the traditional single-component gas test, the sensor of pure SnO_2_ exhibited good selectivity for VOCs and NO at 300 °C and 150 °C, respectively.

In NO, in addition to the above-mentioned effects, Pt also shows a catalytic effect. As a common surface catalyst, Pt has high catalytic activity [45]. Here, it is considered that Pt may catalyze the reaction between NO and VOCs. Taking ethanol as an example, the reaction process is as follows. Firstly, Pt catalyzed the oxidation-reduction reaction between ethanol and NO to produce CO_2_ and NO^−^; then, NO^−^ was decomposed into N_2_ and O^−^. This process increased the amount of O^−^ on the surface. Subsequently, O^−^ further reacted with VOC, as shown in Equations (4) and (5). Therefore, it was observed that the sensitivity of Pt-SnO_2_ to VOCs was greatly improved under the background of 1 ppm NO (air balance). As the presence of Pt has no effect on the adsorption process of NO, SnO_2_ and Pt-SnO_2_ showed similar responses to NO.
(4)$$C_{2}H_{5}\mathrm{OH}+3\mathrm{NO}+{3O}_{2}^{-}\overset{\mathrm{Pt}}{\to }{2\mathrm{CO}}_{2}+\frac{3}{2}N_{2}+{3O}^{-}+{3H}_{2}O$$
(5)$$C_{2}H_{5}{\mathrm{OH}}_{\left(\mathrm{ads}\right)}+{6O}_{\left(\mathrm{ads}\right)}^{-}\to {2\mathrm{CO}}_{2(g)}+{3H}_{2}O_{\left(g\right)}+{6e}^{-}$$

## 4. Conclusions

In summary, SnO_2_ and 1mol% Pt decorated SnO_2_ (Pt-SnO_2_) gas-sensing films were fabricated onto the substrate with prefabricated Pt interdigital electrodes by screen printing. The SnO_2_ sensor had a better response to NO, and the Pt-SnO_2_ sensor had a better response to VOCs. By comparing the response of SnO_2_ and Pt-SnO_2_ to VOCs in an air background and in 1 ppm NO (air balance) under single-component and multi-component atmospheres, it was found that the presence of NO can improve the response of VOCs at low temperature. In addition, the Pt-SnO_2_ sensor showed a significant increase in VOC sensitivity under the NO background. The explanation of this phenomenon is that the noble metal Pt can catalyze the reaction between NO and VOCs to generate more O^−^, which further promotes the adsorption of VOCs. The result of this work is of great significance for the study of gas selectivity. The selectivity cannot be simply defined by a single-component gas test; mutual interference between mixed gases needs to be taken into account.

## Figures and Tables

**Figure 1 nanomaterials-13-00908-f001:**
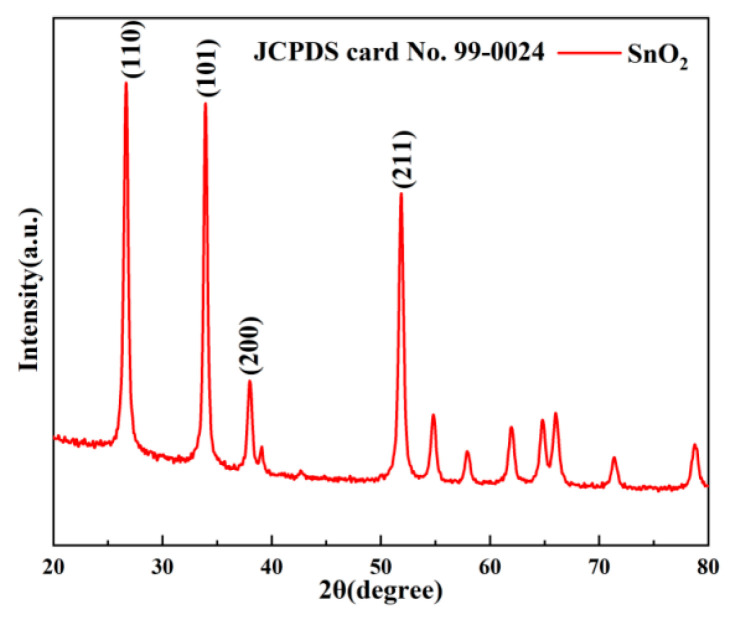
XRD patterns of SnO_2_ nanoparticles.

**Figure 2 nanomaterials-13-00908-f002:**
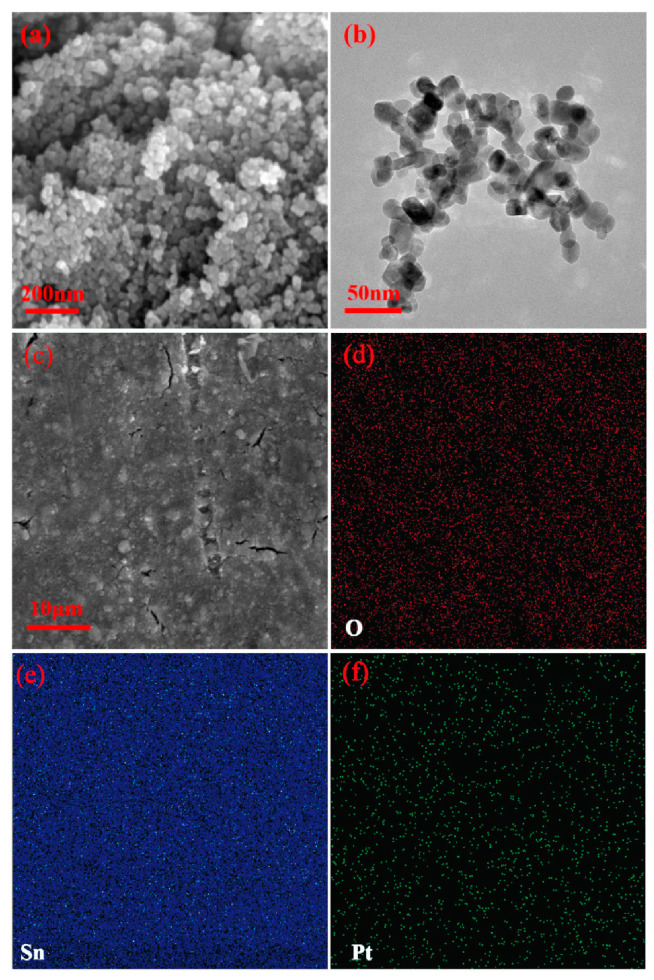
(**a**) SEM and (**b**) TEM images of SnO_2_; (**c**–**f**) EDS mapping of Pt-SnO_2_.

**Figure 3 nanomaterials-13-00908-f003:**
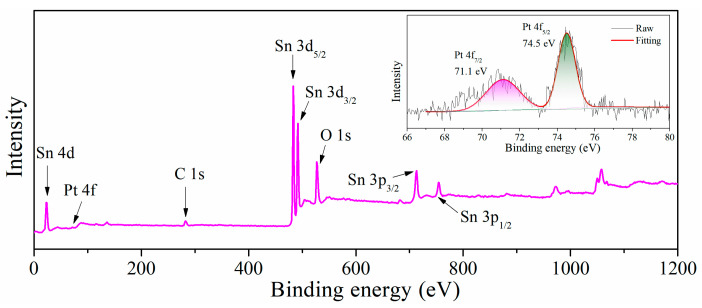
XPS spectra of Pt-SnO_2_.

**Figure 4 nanomaterials-13-00908-f004:**
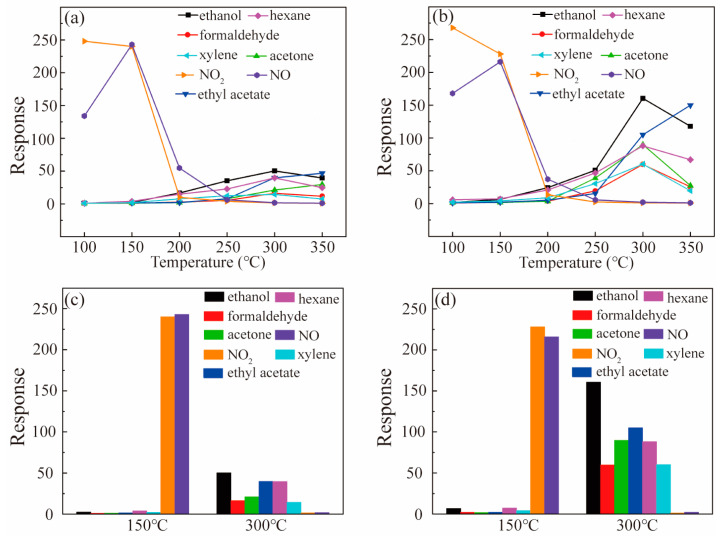
Response curves of (**a**) SnO_2_ and (**b**) Pt-SnO_2_ to different gases in the whole temperature range; response curves of (**c**) SnO_2_ and (**d**) Pt-SnO_2_ to different gases under 150 °C and 300 °C.

**Figure 5 nanomaterials-13-00908-f005:**
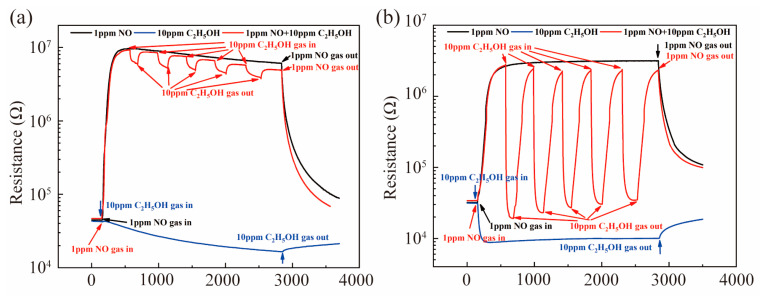
Dynamic resistance curves of (**a**) SnO_2_ and (**b**) Pt-SnO_2_ to 1 ppm NO and 10 ppm C_2_H_5_OH at 150 °C.

**Figure 6 nanomaterials-13-00908-f006:**
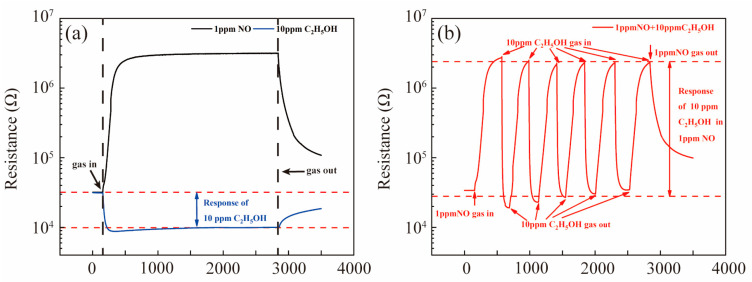
Response of Pt-SnO_2_ to 10 ppm C_2_H_5_OH (**a**) in dry air and (**b**) in 1 ppm NO.

**Figure 7 nanomaterials-13-00908-f007:**
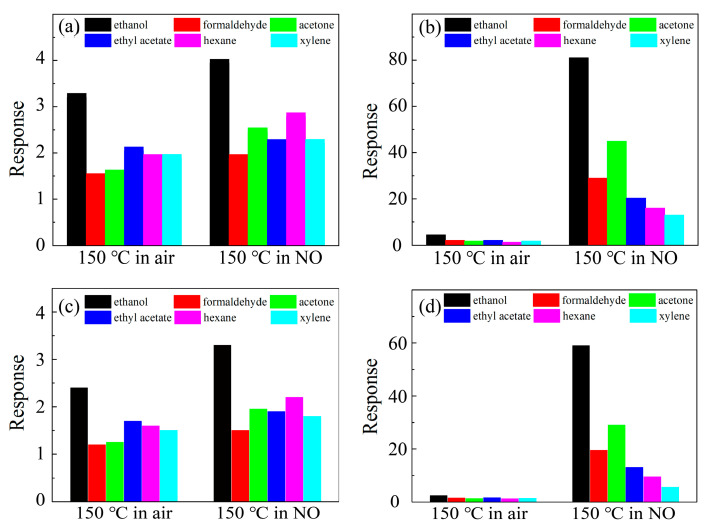
The response curves of SnO_2_ and Pt-SnO_2_ sensors to 10 ppm different VOCs with (**a**,**c**) or without (**b**,**d**) humidity (50% RH) in air and 1 ppm NO.

## Data Availability

Not applicable.

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
