# Peer review of "Cross-Interference of VOCs in SnO2-Based NO Sensors"

_nanomaterials, 2023, doi:10.3390/nano13050908_

Round 1

Reviewer 1 Report

Si et al. presented screen-printed SnO2 NPs on IDE for VOCs and NO sensors. The interference effect /sensors selectivity between VOCs and NO were investigated, improved by the addition of Pt NPs to the SnO2 NPs sensors. The experimental results are interesting, but the presentation of the manuscript can be improved for clarity.

1.       It appears not clear to me that the authors are presenting a VOC or NO sensor or simply to study and compare the interference effect. In the Intro, the authors ‘….conduct a study aimed at the cross interference of VOCs to SnO2-based NO sensor’ …. and then carried out the experiment where ‘gas sensing performance of SnO2 and Pt-SnO2 to different VOCs in air and in 1ppm NO to verify if NO is able to enhance the sensitivity of SnO2 …’ I suggest the authors to rephrase or to clarify the goal of the study.

2.       Incomplete experimental details: Please provide the complete details for materials characterization, ‘homemade organic solvent’ in line 105. What is the dimension or design of the IDE?

3.       In lines 145-146, the synthesized pastes were annealed at 350 oC and 550 oC, but these are not found in the experimental. Are these steps done before the aging of the sensor? Please verify and add the details in the experimental. What is the loading amount of Pt? What is the thickness of the SnO2 film screen-printed on the IDE?

4.       Figure 3:The Pt 4f5/2 (75.5eV) and 4f7/2 (71.8 eV) separation is 3.7 eV (as opposed to 3.3-3.4 eV, with an asymmetrical peak shape). Please verify with reliable resources, and state in the experimental if any fitting or background subtraction was carried out.

5.       Figure 4 has terribly small fonts and is overall unclear and busy. A significant revision is needed there.

6.       Please add the gas in/out labels in Figure 6a, and in Figure 6b, I suppose there is a ‘C2H5OH gas out’?

7.       The authors concluded that the presence of NO can improve the response toward VOCs (which is not seen in Figure 7a for SnO2), and Pt further catalyzes the reaction of NO and VOCs. In Figure 4 at 150 oC, SnO2 showed a higher response to NO than Pt-SnO2, but the 1 ppm of NO was added to VOCs (in Figure 7) has completely no influence on the SnO2 sensor, or they are not visible in the current plots?

8.       The response of Pt-SnO2 sensor toward VOCs at 150 oC with the addition of NO (Figure 7b) followed quite a similar trend to that at 300 oC in air (Figure 4d), it will also be interesting to see if the other conditions follow certain trend. The smaller bars in the histogram are not well visible, the presentation can be improved.

9.       In the abstract and conclusion, I suggest the authors to add which sensor responds better to which type of gas(es) and under which condition(s).

10.   The introduction would profit from some more references to literature on SnO2 based NOx sensors. There are sensors that seem to perform better, see e.g. ACS Appl. Mater. Interfaces 2020, 12, 29, 33386–33396. A comparison (e.g. via table comparing performance) and justification of the current materials and their selection should be made and provided by authors.

Reviewer 2 Report

The manuscript is of interest for the journal but it needs revision. After revision, it can be reconsidered by the journal (for acceptance). Below are comments and suggestions to help the authors improve it:

1\ Line 16:  should be "NO" instead of "nO"

2\ Reference List: Line 281: should be "Sens." instead of "Sensors"

3\ line 285: should be "Interfaces" instead of "Inter."

4\ lines 327, 330: the journal titles should be italicized 

5\ line 333: should be "ACS Sens."

6\ lines 360,355: journal titles must be italicized

7\ Lines 78-80: the sentence should be improved, its English is incorrect

8\ lines 244,245: what is the meaning of "pt" ? please fix !

Reviewer 3 Report

The author introduces the difference in sensitivity and selectivity exhibited by SnO2 and Pt-SnO2 sensors under VOCs and NO combined action. The experiments designed by the authors are comprehensive. The explanation of the mechanism is sufficient and detailed. Only some minor comments need to be addressed.

1.       Figure 2 d to f; EDX mapping figures need to increase color saturation. When printing in 8-bit color, it is difficult to distinguish the colors of elements.

2.       Minor typos need to correct. nO in line 16; “pt” in line 244.

This paper introduces the sensitivity enhancement and selectivity change to NO and VOCs exhibited by SnO-based gas sensors with Pt loading. This article can be classified into the field of Nanomaterial applications. The authors' demonstration and interpretation of the VOCs and NO interactive response phenomenon are the highlights of this article.

The author's English writing is accustomed to using long sentences, which may be an obstacle to readers' understanding.

The experimental design of this article is excellent, the data is sufficient, and the logic is clear.

Citations are appropriate.

Reviewer 4 Report

The authors have provided a study related to SnO2 cross-interferences. The subject is interesting, but there are several critical issues related to the manuscript that must be addressed before the manuscript can be considered for publication:

1.     section 2.1. Materials Synthesis should also refer to the Pt-SnO2, respectively to the impregnation/doping method and to the Pt concentration;  

2.     the sensing properties should be also evaluated in the presence of relative humidity, as background gas that determines cross-interferences;

3.     I ask the authors to explain why they preferred NO in the measurements shown in figures 5-7 and not NO2 as figure 4 would suggest;

4.     the discussion regarding the sensing mechanism was made around O- (eq 1-4) although the measurements were made at 150 and 300 °C. Therefore, I ask the authors to rewrite eq 1-4 according to the working temperature and the presence of NO;

5.     in lines 239-240 the authors state that: “When the temperature is lower than 200℃, O2plays a major role in NO adsorption, the adsorption rate is much higher than the desorption rate”. I ask the authors to support this statement with citations, in the conditions that both oxygen and NO are oxidizing gases;

6.      line 16: nO must be corrected;

7.     I ask the authors to be consistent with the reference to either NO or NOx in the title, abstract and introduction.

Round 2

Reviewer 2 Report

The manuscript was improved, and now it can be accepted

Reviewer 4 Report

I believe that the authors have carefully revised the manuscript and that it is now worth publishing.